# The Impact of COVID-19 on In-Hospital Outcomes of ST-Segment Elevation Myocardial Infarction Patients

**DOI:** 10.3390/jcm10020278

**Published:** 2021-01-14

**Authors:** Sherif Ayad, Rafik Shenouda, Michael Henein

**Affiliations:** 1Faculty of Medicine, Alexandria University, Alexandria 21526, Egypt; 2International Cardiac Center, Alexandria 21526, Egypt; rafikbotros17@hotmail.com; 3Institute of Public Health and Clinical Medicine, Umea University, SE-90187 Umea, Sweden; michael.henein@umu.se

**Keywords:** ST segment elevation myocardial infarction, COVID-19, primary percutaneous intervention

## Abstract

Primary percutaneous coronary intervention (PPCI) is one of the important clinical procedures that have been affected by the COVID-19 pandemic. In this study, we aimed to assess the incidence and impact of COVID-19 on in-hospital clinical outcome of ST elevation myocardial infarction (STEMI) patients managed with PPCI. This observational retrospective study was conducted on consecutive STEMI patients who presented to the International Cardiac Center (ICC) hospital, Alexandria, Egypt between 1 February and 31 October 2020. A group of STEMI patients presented during the same period in 2019 was also assessed (control group) and data was used for comparison. The inclusion criteria were established diagnosis of STEMI requiring PPCI.A total of 634 patients were included in the study. During the COVID-19 period, the number of PPCI procedures was reduced by 25.7% compared with previous year (mean 30.0 ± 4.01 vs. 40.4 ± 5.3 case/month) and the time from first medical contact to Needle (FMC-to-N) was longer (125.0 ± 53.6 vs. 52.6 ± 22.8 min, *p* = 0.001). Also, during COVID-19, the in-hospital mortality was higher (7.4 vs. 4.6%, *p* = 0.036) as was the incidence of re-infarction (12.2 vs. 7.7%, *p* = 0.041) and the need for revascularization (15.9 vs. 10.7%, *p* = 0.046). The incidence of heart failure, stroke, and bleeding was not different between groups, but hospital stay was longer during COVID-19 (6.85 ± 4.22 vs. 3.5 ± 2.3 day, *p* = 0.0025). Conclusion: At the ICC, COVID-19 pandemic contributed significantly to the PPCI management of STEMI patients with decreased number and delayed procedures. COVID-19 was also associated with higher in-hospital mortality, rate of re-infarction, need for revascularization, and longer hospital stay.

## 1. Introduction

Currently, primary percutaneous coronary intervention (PPCI) is the best management strategy for patients presenting with ST-segment elevation myocardial infarction (STEMI) according to the latest guidelines [1]. Studies have shown that time delay in PPCI has negative impact on the clinical outcomes of STEMI patients [2,3].

COVID-19 affected many aspects of human life since its start in early 2020, one of which is prioritizing clinical management of various medical conditions including coronary artery disease, particularly acute coronary syndrome and urgent interventions required for STEMI, a potential life-threatening condition. The WHO classifiesCOVID-19 cases into four categories based on clinical history, presentation, and laboratory findings: confirmed (COVID-19 +), suspected (COVID-19 +/−), contact (COVID-19 C), or non-suspected (COVID-19 NS) [4].

COVID-19 has significantly impacted conventional management of STEMI patients, resulting in practice variabilities between countries. Some countries have changed their reperfusion strategy to fibrinolytic therapy [5,6,7], others still follow the guidelines in performing PPCI to all STEMI patients [8,9,10,11]. The delay in seeking medical advice during the lockdown periods, the time needed for screening for COVID-19 infection, and the fear of healthcare providers regarding cross-infection are the main causes behind the change of practice of managing STEMI patients and the fall in PPCI procedures according to some reports [12,13,14,15].

In this study, we aimed to assess the impact of COVID-19 on in-hospital clinical outcome of STEMI patients managed with PPCI.

## 2. Experimental Section

### 2.1. Study Design

This is a retrospective observational study conducted on consecutive STEMI patients who presented to the International Cardiac Center (ICC) hospital, Alexandria, Egypt between 1 February and 31 October 2020. The inclusion criteria were established diagnosis STEMI (ST segment elevation more than 1 mm in two consecutive leads or new left bundle branch block associated with typical chest pain with or without elevated cardiac markers) fulfilling guidelines recommendation for PPCI treatment [1,16]. The exclusion criteria were previous CABG, cardiogenic shock, previous PCI of the same culprit vessel and severe left main (LM) coronary artery disease. Data from a group of STEMI patients who presented to ICC during the same period of 2019 was used for comparison, as control. Twenty patients in group A and 5 patients in group B were excluded. The study population included 634 patients who were classified into two groups:

Group A:Included 364 STEMI patients treated with PPCI before COVID-19 (year 2019).Group B:Included 270STEMI patients treated with PPCI during COVID-19 (year 2020).

### 2.2. Data Collection

All patients’ demographic data were collected including age, gender, comorbidities (hypertension, diabetes, dyslipidemia), obtained PPCI procedure details including time from symptom onset to first medical contact (FMC), and time from first medical contact to needle (FMC-to-N). From coronary angiograms the following information were collected; the culprit artery, number of diseased vessels, the use of antithrombotic treatment (acetyl salicylic acid, clopidogrel, ticagrelor, heparin, enoxaparin, and glycoprotein IIb/IIIa inhibitors), balloon pre-dilatation, stent details (number, length, and diameter), Thrombolysis In Myocardial Infarction (TIMI) score, flow at the end of the procedure, and duration of hospital stays. Also, any subsequent procedure related complications—e.g., heart failure, stroke, or bleeding—were documented.

### 2.3. Endpoint Measurements

The primary clinical outcomes were the percentage of PPCI procedures performed before and during the COVID-19 and the median time of first medical contact to needle (FMC-to-N), while the secondary outcomes were in-hospital mortality, major adverse cardiac and cerebrovascular events (MACCE) during hospital stay and the duration of hospitalization. MACCE was defined as death, re-infarction, need for revascularization, heart failure, stroke, and bleeding.

### 2.4. Statistical Analysis

Statistical Package for Social Sciences (SPSS version 20.0. IBM Corp, Armonk, NY, USA) was used for data analysis [17]. We described qualitative data using numbers and percentage. For quantitative data we used range (minimum and maximum), mean, standard deviation, and median. Chi-square test was used to compare categorical variables between different groups. Fisher’s exact probability or Monte Carlo correction for Chi-square were used when more than 20% of the cells have expected count less than 5. Mann-Whitney test was used to compare groups for abnormally distributed quantitative variables. A *p*-value of <0.05 was considered significant for all tests.

An informed consent was obtained from every patient or the legal guardians. The study was approved by the local ethics committee (approval number 0304893).

## 3. Results

### 3.1. Patients Characteristics and Number of Procedures

During the COVID-19 period, the number of PPCI procedures was reduced by 25.7% compared with previous year (30.0 ± 4.01 vs.40.4 ± 5.3 case/month). Both patient groups (A and B) were well matched with respect to demographic data and clinical characteristics with no significant difference between them. Only eight patients in group A and five patients in group B were more than 65–70 years of age. The baseline characteristics of both groups are presented in Table 1.

### 3.2. Laboratory Findings

The incidence of lymphopenia was significantly higher in group B than in group A (14.78 ± 5.85 vs. 18.6 ± 6.21, *p* = 0.012), serum ferritin and D-dimer levels were also higher in group B than in group A. Cardiac enzymes, haemoglobin and serum creatinine did not differ between groups. The laboratory findings of both groups are shown in Table 1.

### 3.3. Time FMC-To-N

Patients in group B had significantly longer FMC-to-N compared to patients in group A (125.0 ± 53.6 vs. 52.6 ± 22.8, *p* = 0.001). The FMC-to-N of both groups is presented in Table 1.

### 3.4. Procedural Characteristics of the Two Groups

With regard to the angiographic data, the incidence of multivessel disease was not different between the two groups, as was the culprit artery. Also, the antiplatelet treatment with clopidogrel or ticagrelor did not differ. None of the patients in the two groups received fibrinolytic therapy. All patients in the two groups received drug eluting stents (DES) and no patient had procedure related dissection or perforation. The final TIMI flow at the end of the procedure was similar among patients of both groups. All patients received in-hospital medical treatment and follow up according to the latest STEMI guidelines [1,16]. Data of the procedural characteristics of the studied population are summarized in Table 1.

### 3.5. In-Hospital Outcomes

In hospital mortality was higher in group B (7.4 vs. 4.6%, *p* = 0.036) as was the incidence of re-infarction (12.2 vs. 7.7%) compared to group A, the difference between the two was significant (*p* = 0.041). Twenty patients in group B died, mostly because of arrhythmia (ventricular fibrillation) and the rest developed intractable cardiogenic shock and pulmonary edema. The need for revascularization was also higher in Group B (15.9 vs. 10.7%, *p* = 0.046) but the incidence of heart failure or bleeding was not different between groups. Although there was statistically high stroke prevalence in group A, we are unable to ascertain an exact explanation for it. One possible practice-based explanation for this finding is that in 2019 we used more thrombus aspiration catheters during PPCI than in 2020.The data of procedural outcomes are summarized in Table 2.

### 3.6. Duration of Hospitalization

The duration of hospital stay was significantly longer in group B compared with group A (6.85 ± 4.22 vs. 3.5 ± 2.3 day, *p* = 0.0025).

## 4. Discussion

Findings: COVID-19 pandemic has adversely affected various aspects of health care services including patients with heart disease and acute coronary syndrome [4]. The objective of this study was to evaluate the impact of COVID-19 on STEMI patients requiring conventional PPCI treatment and their clinical outcomes. Our results show that all studied STEMI patients were treated with PPCI without need for fibrinolytic therapy, even for highly suspected COVID-19 patients. However, the frequency of PPCI treatment was significantly reduced and the intervention was delayed when compared with 2019 controls. Also, the hospital stay was prolonged and associated with some complications including re-infarction, need for coronary artery bypass surgery, CVS and increased in-hospital mortality. Despite that, the prevalence of developed heart failure and bleeding was not different from controls, treated by similar strategy a year before COVID-19.

Comparative results: Our findings can be summarized in showing significant change in STEMI practice during COVID-19 with delayed acute presentation and its management. The delay in presentation was mainly due to patients’ fear of catching the viral infection at the hospital. This finding in ICC is compatible with other countries. HunShing Kwok et al. reported a dramatic reduction in PPCI procedures in the UK during the lockdown period [18], Dingcheng Xiang et al. reported 62% less PPCI in China [19], and 73 centers reported 40% reduction in PPCI in Spain [20].The delayed PPCI was merely due to the screening tests performed before procedure, particularly in highly suspected patients who occasionally required other necessary investigations first, e.g., chest computed tomography (CT) scans. The increased in-hospital mortality with COVID-19 is similar to that reported by Dingcheng Xiang et al. [19] but contradicted Hun Shing Kwok et al. reports [18]. Other important findings in our study were the increased rate of re-infarction, the need for revascularization and the doubled hospital stay period, during the pandemic despite similar incidence of heart failure, stroke and bleeding. These findings were similar to that reported by Dingcheng Xiang et al. [19] but contradicted Hun Shing Kwok et al. results [18] which reported significant reduction ofin-hospital stay period.

It seems therefore that the internationally agreed impact of COVID-19 on conventional interventional management of STEMI is mainly during the acute phase of the disease with delayed presentation, reduced number of cases, and delayed procedure. While the former is mainly patient related, the latter is hospital controlled which is based on the nature of presentation of individual patients. In this scenario, it cannot be ignored that the rest of the clinical outcome is determined by the extent of co-morbidities, and severity of COVID-19 infection which vary between individual patients.

Limitations: This study has some obvious limitations. The recruited patients were those referred to the ICC hospital with STEMI diagnosis, mostly by individual cardiologists or other local hospitals, thus do not reflect a population. The follow-up duration was short and concerned only in-hospital stay, based on the study nature and design. Although patients were referred from different sources, they were all managed in one center from which the results were generated.

## 5. Conclusions

Our study shows that COVID-19 was associated with a significant decrease in the number of STEMI patients treated by PPCI at the ICC- Egypt, delayed procedure, higher in-hospital mortality, higher rates of re-infarction, need for repeat revascularization, and longer duration of hospital stay but with similar rates of heart failure, stroke, and bleeding.

## Figures and Tables

**Table 1 jcm-10-00278-t001:** Baseline characteristics, laboratory findings, procedural characteristics of the studied populations.

	Group A *n* = 364		Group B *n* = 270		*p*-Value
**Age**					
Range	36–88		35–82		0.568
Mean ± S.D.	58.9 ± 13.35		57.1 ± 12.60		
**Gender**					
Male	312	85.7%	220	81.5%	0.607
Female	52	14.3%	50	18.5%	
**Risk factors**					
Diabetes mellitus	130	35.7%	95	35.2%	0.521
Hypertension	156	42.9%	107	39.6%	0.411
Dyslipidemia	182	50.0%	122	45.2%	0.501
Smoking	208	57.1%	123	45.6%	0.364
**Troponin**					
Range	0.003–8.68		0.01–10.0		0.078
Mean ± S.D.	1.07 ± 2.21		1.65 ± 2.62		
**CKmb**					
Range	1.27–261.9		1.32–270.0		0.105
Mean ± S.D.	115.94 ± 76.29		124.3 ± 58.9		
**Haemoglobin**					
Range	9.3–17.1		9.5–16.0		0.524
Mean ± S.D.	13.87 ± 1.85		13.9 ± 1.71		
**Lymphocytes**					
Range	12–36		8–25		0.012 *
Mean ± S.D.	18.6 ± 6.21		14.78 ± 5.85		
**D dimer**					
Range	130–500		152–1500		0.0031 *
Mean ± S.D.	302.0 ± 132.17		505.6 ± 201.3		
**Serum ferritin**					
Range	72.0–135.0		85.0–166.0		0.011 *
Mean ± S.D.	93.48 ± 39.8		118.5 ± 42.51		
**Serum creatinine**					
Range	0.59–4.03		0.60–3.52		0.211
Mean ± S.D.	1.12 ± 0.66		1.26 ± 0.71		
**FMC-to-N (min)**					
Range	15–85		60.0–280		0.001 *
Mean ± S.D.	52.6 ± 22.8		125.0 ± 53.6		
	No.	%	No.	%	
**MVD** **SVD**	43 321	11.8 88.2	63 207	23.3 76.7	0.389
**Culprit vessel**					
LAD	216	59.3	128	47.4	0.089
RCA	108	29.7	97	35.9	
LCX	40	11	45	16.7	
Clopidogrel Ticagrelor	221 143	60.7 39.3	155 115	57.3 42.7	0.410

*p* value for comparing between the two studied groups. *: Statistically significant at *p* ≤ 0.05.

**Table 2 jcm-10-00278-t002:** In hospital outcomes of the studied population

	Group A *n* = 364		Group B *n* = 270		*p*-Value
	No.	%	No.	%	
In-hospital mortality	17	4.6	20	7.4	0.036 *
Re-infarction	28	7.7	33	12.2	0.041 *
Need for revascularization	39	10.7	43	15.9	0.046 *
Heart Failure	117	32.1	96	35.6	0.258
CVS	20	5.5	10	3.7	0.022
Bleeding	39	10.7	30	11.1	0.511

*p* value for comparing between the two studied groups. *: Statistically significant at *p* ≤ 0.05.

## Data Availability

Data available in a publicly accessible repository.

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
