# Peer review of "The Impact of COVID-19 on In-Hospital Outcomes of ST-Segment Elevation Myocardial Infarction Patients"

_jcm, 2021, doi:10.3390/jcm10020278_

Round 1

Reviewer 1 Report

The manuscript is interesting and well written. I have some suggestions.

The authors wrote “incidence of mortality was higher in group B… and the necessity of revascularization was also higher in Group B”. Although group A and B are matched for age, it would be very interesting to know if in group there are patients older (>65-70 years).

There are too many Tables. Table 1, table 2, table 3, table 4 should be merged. results of Table 6 should be inserted in the text. One table for one parameter is not the better solution to show.

Author Response

Response to Reviewer 1 Comments

Point 1: The authors wrote “incidence of mortality was higher in group B… and the necessity of revascularization was also higher in Group B”. Although group A and B are matched for age, it would be very interesting to know if in group there are patients older (>65-70 years).

Response 1: Thank you for the comment. Only 8 patients in group A and 5 patients in group B were more 65-70 years of age. This information has now been added to the text.

Point 2: There are too many Tables. Table 1, table 2, table 3, table 4 should be merged. results of Table 6 should be inserted in the text. One table for one parameter is not the better solution to show.

Response 2: Thank you for the suggestion Table 1, 2, 3 and 4 have now been merged in one table, table 6 is already inserted in the text, and the table is removed.  

Reviewer 2 Report

This is a retrospective study evaluating the impact of COVID-19 pandemic on management of STEMI patients. During the COVID-19 pandemic, the number of primary PCI for STEMI was significantly reduced, and a higher in-hospital mortality and a higher incidence of re-infarction were observed. Although the impact of COVID-19 on STEMI is an important theme, there are several issues that must be properly addressed.

  1. In terms of a study design, inclusion and exclusion criteria are ambiguous. Only the patients who underwent PCI were enrolled? How many patients underwent thrombolytic therapy? Why were the patients with cardiogenic shock or LMT disease were excluded?
    Furthermore, flow chart should be added with the number of excluded patients.
  2. It is difficult to generalize the results of this paper due to too small number of patients from one hospital.
  3. The difference of in-hospital death might be a main message of this paper. Nevertheless, there was no mention about the cause of death.

  1. The majority of the study population is patients transported from nearby hospitals. Patients are being transported where it is determined that performing PCI will have beneficial results, and this is subject to a very crucial selection bias.

  1. How many patients received thrombolytic therapy before PCI?

  1. Based on the results of this study, can we say that prognosis remains the same regardless of the timing of PCI within 24 hours f AHJ2020 STEMI presenting >12h or STEMI patients?

Author Response

Response to Reviewer 2 Comments

Point 1: In terms of a study design, inclusion and exclusion criteria are ambiguous. Only the patients who underwent PCI were enrolled? How many patients underwent thrombolytic therapy? Why were the patients with cardiogenic shock or LMT disease were excluded?

Response 1: Thank you for your comments. The objective of this study was to assess the impact of COVID 19 on the intervention procedure strategy and clinical outcome, hence patients who underwent thrombolytic therapy were not included in the study. Likewise, patients with cardiogenic shock were not included since not all of them were treated by PCI, some of them were treated with emergency CABG thus not fulfilling the study design. These two groups are known for their individualistic clinical treatment approach, rather than follow timely plan as is the case with PCI.

Furthermore, flow chart should be added with the number of excluded patients.

Thank you. 20 patients in group A and 5 patients in group B were excluded. This information has now been added to the revised draft.

Point 2: It is difficult to generalize the results of this paper due to too small number of patients from one hospital.

Response 2: We agree with the Reviewer’s comment that it is difficult to generalize the results of this paper because of the small number of patients. This had been stated in the limitation section.

Point 3: The difference of in-hospital death might be a main message of this paper. Nevertheless, there was no mention about the cause of death.

Response 3: Twenty patients in group B died, mostly because of arrhythmia (ventricular fibrillation) and the rest developed intractable cardiogenic shock and pulmonary oedema. This information has now been added to the text.

Point 4: The majority of the study population is patients transported from nearby hospitals. Patients are being transported where it is determined that performing PCI will have beneficial results, and this is subject to a very crucial selection bias.

Response 4: We agree, that the studied group of patients represent a potential bias however, they were uniformly enrolled having all fulfilled the study inclusion criteria.

Point 5: How many patients received thrombolytic therapy before PCI?

Response 5: None of the patients received fibrinolytic therapy in the two groups of patients.

Point 6: Based on the results of this study, can we say that prognosis remains the same regardless of the timing of PCI within 24 hours f AHJ2020 STEMI presenting >12h or STEMI patients?

Response 6: Prognosis was not the same since Group B carried higher mortality, re-infarction and higher need for revascularization.

Reviewer 3 Report

T​his study compared STEMI presentation during the COVID19 pandemic to a period of time pre-pandemic. They noted during COVID19, there were 25% fewer STEMIs, time to needle was significantly increased, and there was high in-hospital mortality, re-infarction, and revascularization. I have a number of comments:   -There are multiple spacing, grammatical errors. I would encourage review of the paper by a fluent English speaker.
-The statement "While fibrinolytic therapy represents the preferred repursion therapy" is debatable.
-Why aren't all STEMI patients during the pandemic treated as suspected COVID +? This would mean wearing appropriate personal protective equipment and thus minimizing delay during PCI? -Along the same lines,  did the patients in Group B test positive for COVID? This data is important to show.
-The authors do not detail why there was a delay in PCI for these patients? Was it awaiting a COVID test? Why would a chest CT scan be done in STEMI patients (I can understand a CT angiogram if a dissection was suspected) but why a chest CT?
-Why do the authors suggest there was more stroke in Group A?
-What was the cause of in-hospital mortality? If these patients were COVID+ this could play a role in higher mortality in Group B. -What is unique about this study compared to multiple others that have shown similar trends in PCI during COVID?

Author Response

Response to Reviewer 3 Comments

Point 1: T​his study compared STEMI presentation during the COVID19 pandemic to a period of time pre-pandemic. They noted during COVID19, there were 25% fewer STEMIs, time to needle was significantly increased, and there was high in-hospital mortality, re-infarction, and revascularization. I have a number of comments:   -There are multiple spacing, grammatical errors. I would encourage review of the paper by a fluent English speaker.

Response 1: Thank you. We have checked the manuscript by a fluent English speaker.

Point 2: The statement "While fibrinolytic therapy represents the preferred repursion therapy" is debatable.

Response 2: We have deleted this statement and rephrased the paragraph.

Point 3: Why aren't all STEMI patients during the pandemic treated as suspected COVID +? This would mean wearing appropriate personal protective equipment and thus minimizing delay during PCI? -Along the same lines,  did the patients in Group B test positive for COVID? This data is important to show.

Response 3: Thank you. Indeed, all health medical personal were wearing personal protective equipment. Patients were all tested for COVID before PPCI according to the Egyptian Health Ministry protocol. All suspected or positive COVID patients were transferred to specific isolation hospitals after receiving reperfusion therapy.PCR was not available at the ICC but only at the specific isolation hospitals.

Point 4: The authors do not detail why there was a delay in PCI for these patients? Was it awaiting a COVID test? Why would a chest CT scan be done in STEMI patients (I can understand a CT angiogram if a dissection was suspected) but why a chest CT?

Response 4: Fast COVID testing is not available in EGYPT. Even PCR tests are limited in number and the results of which are available after 24 hours. Thus, the diagnosis is mainly based on fever, lymphopenia, elevated CRP and D-Dimer and lung patches on CT chest .Only suspected patients had the PCR test after being transferred to specific isolation hospitals.

Point 5: Why do the authors suggest there was more stroke in Group A?

Response 5: We did not suggest that but stated that similar rates of stroke, heart failure and bleeding.

Point 6: What was the cause of in-hospital mortality? If these patients were COVID+ this could play a role in higher mortality in Group B

Response 6: We didn’t have sure documentation that deaths are among COVID positive patients as we did not perform the PCR test in our hospital. Twenty patients in group B died, mostly because of arrhythmia (ventricular fibrillation) and the rest developed intractable cardiogenic shock and pulmonary oedema. This information has now been added to the text.

Point 7: What is unique about this study compared to multiple others that have shown similar trends in PCI during COVID?

Response 7: The unique observation in this study is the use of the conventional reperfusion treatment strategy in COVID patients. We did not follow other centers recommendations in using fibrinolytic therapy e.g. China and Europe but continued to perform PPCI for all STEMI patients. To our knowledge this is the first report addressing the impact of COVID 19 on PPCI from Alexandria, EGYPT.

Round 2

Reviewer 2 Report

nothing particular

Author Response

Thank you.

Reviewer 3 Report

The rate of stroke reported between group A and group B is different according to the p value of 0.022. The comments about this need to be modified.

Author Response

Response to Reviewer 3 Comment

Point: The rate of stroke reported between group A and group B is different according to the p value of 0.022. The comments about this need to be modified.

Response: Thank you. Although there was statistically high stroke prevalence in group A, we are unable to ascertain an exact explanation for it. One possible practice-based explanation for this finding is that in 2019 we used more thrombus aspiration catheters during PPCI than in 2020.

We added this in lines 123,124 and 125 of the manuscript.
